# World Health Organization Danger Signs to predict bacterial sepsis in young infants: A pragmatic cohort study

Omolabake Akinseye[1‡], Constantin R. Popescu[2,3‡], Msandeni Chiume-Kayuni[4,5], Michael A. Irvine[6,7], Norman Lufesi[8], Tisungane Mvalo[9,10], Niranjan Kissoon[1,2], Matthew O. Wiens[11], Pascal M. Lavoie[1,2]*

1 Department of Pediatrics, Faculty of Medicine, University of British Columbia, Vancouver, Canada, 2 British Columbia Children's Hospital Research Institute, Vancouver, Canada, 3 Department of Pediatrics, Université Laval, Québec, Canada, 4 Department of Pediatrics, Kamuzu Central Hospital, Lilongwe, Malawi, 5 Kamuzu University of Health Sciences, Lilongwe, Malawi, 6 British Columbia Centre for Disease Control, Vancouver, Canada, 7 Faculty of Health Sciences, Simon Fraser University, Burnaby, Canada, 8 Department of Curative and Medical Rehabilitation, Ministry of Health, Lilongwe, Malawi, 9 University of North Carolina Project Malawi, Lilongwe, Malawi, 10 Department of Pediatrics, University of North Carolina at Chapel Hill School of Medicine, Chapel Hill, North Carolina, United States of America, 11 Department of Anesthesiology, Pharmacology & Therapeutics, Faculty of Medicine, University of British Columbia, Vancouver, Canada

‡ OA and CRP contributed equally to this work as co-first authors.
* plavoie@bcchr.ca

**Data Availability Statement:** All data can be found in the manuscript and supporting information files.

## Abstract

Bacterial sepsis is generally a major concern in ill infants. To help triaging decisions by front-line health workers in these situations, the World Health Organization (WHO) has developed danger signs (DS). The objective of this study was to evaluate the extent to which nine DS predict bacterial sepsis in young infants presenting with suspected sepsis in a low-income country setting. The study pragmatically evaluated nine DS in infants younger than 3 months with suspected sepsis in a regional hospital in Lilongwe, Malawi, between June 2018 and April 2020. Main outcomes were positive blood or cerebrospinal fluid (CSF) cultures for neonatal pathogens, and mortality. Among 401 infants (gestational age [mean ± SD]: 37.1±3.3 weeks, birth weight 2865±785 grams), 41 had positive blood or CSF cultures for a neonatal pathogen. In-hospital mortality occurred in 9.7% of infants overall (N = 39/401), of which 61.5% (24/39) occurred within 48 hours of admission. Mortality was higher in infants with bacterial sepsis compared to other infants (22.0% [9/41] versus 8.3% [30/360]; p = 0.005). All DS were associated with mortality except for temperature instability and tachypnea, whereas none of the DS were significantly associated with bacterial sepsis, except for "unable to feed" (OR 2.25; 95%CI: 1.17–4.44; p = 0.017). The number of DS predicted mortality (OR: 1.75; 95%CI: 1.43–2.17; p<0.001; AUC: 0.756), but was marginally associated with positive cultures with a neonatal pathogen (OR 1.22; 95%CI: 1.00–1.49; p = 0.046; AUC: 0.743). The association between number of DS and mortality remained significant after adjusting for admission weight, the only statistically significant co-variable (OR 1.75 [95% CI: 1.39–2.23]; p<0.001). Considering all positive cultures including potential bacterial contaminants resulted a non-significant association between number of DS and sepsis (OR 1.09 [95% CI: 0.93–1.28]; p = 0.273). In conclusion, this study shows that DS were strongly

**Funding:** This study was support by a grant from Grand Challenges Canada (R-ST-POC-1707-04630). PML was supported by the BC Children's Hospital Research Foundation through the Investigator Grant Award Program. CRP was funded by a Sue Carruthers Graduate Studentship from the BC Children's Hospital Foundation. TM also received a Grand Challenges Exploration grant from the Bill and Melinda Gates Foundation (OPP1211936). The funders had no role in study design, data collection and analysis, decision to publish, or preparation of the manuscript.

**Competing interests:** The authors have declared that no competing interests exist.

associated with death, but were marginally associated with culture-positive pathogen sepsis in a regional hospital setting. These data imply that the incidence of bacterial sepsis and attributable mortality in infants in LMIC settings may be inaccurately estimated based on clinical signs alone.

## Introduction

Three million children die each year worldwide in their first month of age, with a significant proportion of these deaths occurring during the first week [1]. Severe infections (including sepsis, pneumonia, diarrhea and viral diseases) are common causes of serious illnesses in young infants [2]. Up to 25% of cases in low- and middle-income countries (LMICs) have identified bacterial causes [3]. However, the lack of diagnostic resources, particularly in LMICs makes it difficult to accurately quantify this burden. This gap has enormous implications in terms of implementing policies and guidelines for the use of antimicrobials to prevent these deaths [1]. At the individual level, identifying bacterial causes for sepsis is crucial to prevent fatalities and guide the correct empiric antibiotic therapy in the context of rising antimicrobial resistance. Indeed, the misuse, or even the overuse, of antibiotics carries a risk of subsequent treatment failure due to colonization by resistant bacteria [4, 5].

To assist front-line health workers, the World Health Organization (WHO) and United Nations International Children's Emergency Fund developed algorithms in the early 1990s to help triage interventions for sick children [6]. This Integrated Management of Childhood Illness (IMCI) strategy included the identification of clinical signs to alert the need for escalation of care, later termed Danger Signs (DS) [6]. Subsequently, the Young Infant Clinical Signs Study Group carried out a multi-centered study in six LMICs and found that seven signs (i.e., history of difficulty feeding, grunting, convulsions, movement only when stimulated, respiratory rate ≥60 breaths per minute (bpm), severe chest indrawing, temperature below 35.5˚C or above 37.5˚C, prolonged capillary refill, cyanosis and stiff limbs) best predicted the need for higher level of care, with a sensitivity of 74% to 85%, and specificity of 75% to 79%, depending on infants' age (i.e., 0 to 6 days versus 7 to 59 days old) [7]. A systematic review of five studies on 17,506 infants found similar results [8].

Over time, the use of DS has been extended to provide a reasonable basis for the initiation of empiric antibiotic treatment [9]. However, these signs have neither been developed nor validated to diagnose sepsis of bacterial etiology. Neonatal disorders such as transient tachypnea of the newborn, hypoglycemia, hyaline membrane disease, birth asphyxia, and viral infections (to mention only a few), present like bacterial sepsis [10]. Hence, a key question is whether DS can reasonably identify young infants who truly benefit from antibiotic treatment. Only a few studies have attempted to answer this question, but certainly none that have, to the best of our knowledge, been conducted prospectively, in a pragmatic setting where clinicians, not research staff, pragmatically reported DS [11].

The objective of this study was to determine the extent to which WHO DS predict bacterial sepsis in young infants, in a LMIC hospital setting where front-line health workers were asked to assess and record the presence of DS in infants who presented with suspected sepsis.

## Materials and methods

### Study approval

The study was approved by the National Health Science Research Committee of Malawi (#17/8/1819) and the University of British Columbia's Children's and Women's Research Ethics

Board (#H16-02639). The study was supported by the chair of the Department of Pediatrics at Kamuzu Central Hospital and discussed with leaders from the Pediatric and Child Health Association (PACHA), UNICEF Malawi and the Ministry of Health of Malawi.

## Study design, setting & population

This study was a pragmatic cohort study, and secondary analysis of data from a parent study that aimed to identify molecular RNA-sequencing signatures of bacterial sepsis in young infants with suspected sepsis [12]. In the parent study, infants below 3 months of age with suspected sepsis, defined according to the attending staff's clinically suspicion, were sequentially enrolled "around the clock" at the time they presented to the neonatal or pediatric units at Kamuzu Central Hospital in Lilongwe, Malawi between June 5th, 2018, and April 6th, 2020 (including recruitment, exposure and data collection). To minimize bias towards false-negative blood culture, infants were excluded if they had been on antibiotics more than 4 hours prior to the cultures. About 851 infants presented with suspected sepsis and the vast majority were not enrolled because staff were unable to get consent–either parents declined, or were not approached for consent due to lack of time and/or competing clinical demands. Written informed consent was obtained from parents/guardian of each participant, in English or Chichewa (main local language). Enrolled infants were followed during hospitalization at Kamuzu Central Hospital, until transferred to another facility or discharged home.

The Kamuzu Central Hospital the largest referral hospital in central Malawi, where approximatively 3,000 infants are admitted to its neonatal unit annually. "Suspected sepsis" was defined as per the clinical staff, essentially in an infant that looked unwell where bacterial sepsis could not be excluded clinically. Before the study, the staff at Kamuzu Central Hospital did not have routinely access to blood culture or other ancillary tests, IV fluids and gavage feeds were also most often not available due to limited resources and supplies. Available interventions include antibiotics, assistance with feeding and routine surgeries when necessary. Oxygen and continuous positive pressure ventilation were only available to an extremely limited number of infants, on a limited basis when the hospital had electrical power. Invasive mechanical ventilation was not possible. During the study, investigators did not have access to information that could identify participants, except for the consent forms which were kept under the responsibility of the local study PI (MCK). The study was administered and local research staff were hired through a partnership with the Parent and Child Health Initiative (PACHI), a Malawian non-profit non-governmental organization, which works in collaboration with the Ministry of Health.

## Clinical data and blood samples collection

Each infant had an initial clinical evaluation by a healthcare worker (or staff, either a nurse or clinical officer). The presence of the following nine DS were prospectively collected at presentation, using electronic forms, on a tablet device: i) history of not feeding well (interviewing the parent/caregiver), entered as "feeding well", "unable to feed" or "uninterested in feeding" (combining the latter two categories), ii) fast breathing ≥60 bpm (with respiratory rate visually counted by attended clinician), iii) temperature instability, defined as axillary body temperature <35.5°C or >38°C, (measured using an electronic Welch Allyn SureTemp™ thermometer, model 692), iv) drowsiness/difficulty to wake, v) grunting, vi) severe chest recession, vii) central cyanosis, viii) movements only when stimulated or no movement at all and ix) convulsions (defined pragmatically by attending clinician). Additionally, clinical staff had to report whether they felt the infant was "well", "ill" or "severely ill" at the time of assessment. However, in keeping with a pragmatic design, the study did not attempt to enforce strict criteria for

illness status, or for to define suspected sepsis or the presence or absence of DS. All infants also had a standardized aerobic blood culture (minimum 2 mL of blood) collected within 4 hours of presentation, using a rigorously implemented aseptic protocol (with in-training of all front-line healthcare providers prior to study initiation), as detailed [13]. Cerebrospinal fluid (CSF) cultures were collected by clinical staff, whenever they felt it was clinically indicated, at their discretion, based on availability of supplies, and the cost of the test that had to be paid by parents. No urine cultures were performed.

Blood cultures, were paid by the study, and processed into BACTEC PEDS Plus culture bottles incubated for 5 days in a BD BACTEC 9050 instrument. Identification of organisms from blood or CSF cultures was completed in-house using bioMerieux API, and BD Crystal kits at a local University of North Carolina-affiliated research facility, as described [12]. Bacterial isolates that could be restored (∼60% of isolates) were also analyzed for further confirmation of identity on a Microflex LT MALDI-TOF mass spectrometer, using the BioTyper software version 3.1 (Bruker Daltonik, Germany). Culture results were made available to clinicians involved in the care of the infants.

## Outcome measures

The primary outcomes of interest were *mortality*, and *culture-positive sepsis* defined as the bacterial growth of a known neonatal pathogen in blood or CSF culture sample. Micro-organisms not commonly considered as pathogens in term or near-term infants, such as *coagulase-negative Staphylococcus*, *Micrococcus*, *Corynebacterium*, *Bacillus species*, or streptococcus species such as *S. oralis* were considered as potential contaminants and were excluded from the main bacterial sepsis outcome [14], but these potential contaminants were included in a sensitivity analysis using an alternate bacterial sepsis outcome "including potential contaminants" (**S1 Table**). In-hospital mortality was recorded until discharge home or transfer to another hospital facility. Data were considered missing when infants were transferred within 72 hours after initial admission to the hospital.

## Statistical analysis

In the parent study, the sample size was set on the original molecular signature aim, at 500 infants over two years, assuming a 20% incidence of bacterial sepsis and expecting to identify between 10 and 50 differentially expressed molecular gene markers [12]. However, the study needed to be stopped early, in April 2020 due to the COVID-19 pandemic, so in the end only 401 infants were enrolled. All infants originally enrolled in the parent study were analyzed. Demographic characteristics were reported descriptively using the mean/median with dispersion. The nine aforementioned WHO DS were used as predictors. All quantitative variables were analyzed as such, except for fast breathing and temperature instability which were categorized according to WHO definitions (see above). Odds ratios were calculated for each DS and the main outcomes of bacterial sepsis or mortality. Logistic regressions were conducted between the number of DS and the main outcomes. In separate analyses, odds ratios were calculated using logistic regressions between mean DS (i.e., the number of DS divided by the available DS data) to determine the effect of missing DS data. Unadjusted logistic regression models, including demographic variables, were conducted to determine the predictive value of the DS. The predictive performance of DS for each outcome was assessed using the area under receiver operator characteristic curve (AUC-ROC). Sociodemographic variables associated with mortality, as well as their independent effects were analyzed using regression models. Only complete cases were used in the analysis for associations with individual DS, or the outcome "cumulative number of DS". However, analyses were also conducted using "mean

number of DS", to account for missing values, with no significant differences in results. Statistical significance was considered at p-value <0.05 with no adjustment made for multiple testing in unadjusted models. Analyses were performed using R (version 4.1.0) and SPSS version 20 (IBM, Armonk, NY, USA). The complete dataset for this study can be found in **S1 Dataset**.

### Inclusivity in global research

Additional information regarding the ethical, cultural, and scientific considerations specific to inclusivity in global research is included in the **S1 Questionnaire**.

## Results

### Study population

The majority of the 401 infants included in the study were term or late preterm infants, born vaginally, presented from home in their first week of post-natal life, and had roughly equal female/male distribution (**Table 1**). Most infants were reported by clinicians to look ill at the time of initial presentation (69.3%; 278/401), 22.2% (89/401) were severely ill, and 8.5% appeared well (34/401). In total, 79.8% (320/401) presented at least one DS. Overall, 56.9%

**Table 1. Socio-demographic characteristics of the study population.**

| Participant characteristics | | N with data |
|---|---|---|
| Gestational age at birth (weeks), mean (SD) | 37.1 (3.3) | 397 |
| Birth weight (gram), mean (SD) | 2865 (785) | 380 |
| Female, no (%) | 185 (46) | 401 |
| Admitted from, no (%) | | 401 |
| Inborn (i.e., admitted from birth hospital) | 161 (40.1) | |
| Transferred from another facility | 224 (55.8) | |
| Home | 16 (4.0) | |
| Post-natal age at presentation (days), median (IQR) | 4 (2–16) | 401 |
| Mode of delivery, no (%) | | 398 |
| Caesarean section | 101 (25.2) | |
| Vaginal delivery | 297 (71.8) | |
| Duration of illness before presentation to hospital (hours), median (IQR) | 24 (12–48) | 341 |
| Distance of travel to hospital (km), median (IQR) | 17 (11–25) | 342 |
| Illness status, number on infants (%) | | 401 |
| Well | 34 (8.5) | |
| Ill | 278 (69.3) | |
| Severely ill | 89 (22.2) | |
| Total duration of antibiotics (days), median (IQR) | 6 (5–9) | 401 |
| Duration of hospital stay (days), median (IQR) | 7 (5–9) | 401 |
| Danger Signs, number of infants (%) | | |
| Not feeding well | 171 (43.3) | 395 |
| Convulsions | 28 (7.0) | 401 |
| Drowsy / Difficult to wake | 33 (8.2) | 401 |
| Movements only when stimulated or no movements | 26 (6.5) | 401 |
| Fast breathing > = 60 per min | 104 (27.1) | 384 |
| Grunting | 30 (7.5) | 401 |
| Severe chest recessions | 51 (12.7) | 401 |
| Temperature instability >38˚C or <35.5˚C | 118 (32.7) | 361 |
| Central cyanosis | 29 (7.2) | 401 |

(228/401) of infants had been ill for at least 24 hours prior to initial presentation, and their parents had traveled a median distance of 17 kilometers, often by foot or taxi, to reach the nearest hospital. In total, 94/401 (23.4%) infants had received antibiotics prior to the blood culture, for a median of 2 hours (range 0.75 to 3.5 hours). The duration of antibiotic use in infants ranged from 1 to 30 days (**Table 1**). Few infants with negative blood culture had antibiotics discontinued after 48 hours (21/401, 5.2%).

## Outcome data

The proportion of infants with bacterial sepsis, considering only known pathogens in term and near-term infants, was 10.2% (41/401). Blood (39/401) and CSF (2/204) cultures were positive for pathogens in 9.7% and 1.0% of the infants, respectively. Of 230 infants under 7 days old, 21 (9.1%) had bacterial sepsis with a pathogen versus 20 of the 171 infants (11.7%; $p = 0.402$ by chi-square test) $\geq$ 7 days old. The proportion of gram-positive sepsis (21/39) and gram-negative sepsis (20/39) were similar among all infants (**S1 Table**), and were also similar between infants under 7 days (9 versus 12) and infants $\geq$ 7 days old (12 versus 8; $p = 0.272$ by chi-square test, noting however the small sample size for comparison). The proportion of sepsis was also similar in inborn infants (19/161) versus infants transferred from another hospital facility (22/224) or admitted from home (0/16). Three infants were transferred before 72 hours from admission, so the mortality outcome was considered missing in these infants. Among the remaining infants, mortality was 9.7% (i.e., 39/398), of which 61.5% [24/39] died within 48 hours.

## Associations with mortality and culture-positive outcomes

The following DS were significantly associated with mortality (**Table 2**): *not feeding well*, *convulsions*, *drowsy/difficult to wake*, *movements only when stimulated or no movements*, *grunting*, *severe chest recessions*, *central cyanosis*. In contrast, *fast breathing* or *temperature instability* were not significantly associated with mortality. The number of DS was also significantly associated with mortality (OR 1.75 [95%CI: 1.43–2.17] per cumulative DS; $p<0.001$; AUC: 0.743) (**S1 Fig**). Admission weight was significantly associated with mortality, but neither age of the mother, gestational age, birth weight nor sex were significantly associated with mortality (**S2 Table**). When adjusting for these co-variables, both the cumulative (OR 1.75 [95% CI: 1.39–2.23]; $p<0.001$) and mean number of DS (1.80 [95% CI: 1.44–2.30]; $p<0.001$) remained highly significantly associated with mortality (**S2 Table**).

**Table 2. Associations between DS and mortality.**

| Variable | Odds ratio (OR) | 95%CI | P value |
|---|---|---|---|
| Not feeding well | 7.21 | 3.27–18.2 | <0.001* |
| Convulsions | 3.52 | 1.31–8.59 | 0.008* |
| Drowsy / Difficult to wake | 5.04 | 2.12–11.4 | <0.001* |
| Movements only when stimulated or no movements | 4.89 | 1.88–11.9 | <0.001* |
| Fast breathing > = 60 per min | 1.86 | 0.87–3.85 | 0.100 |
| Grunting | 3.95 | 1.54–9.34 | 0.002* |
| Severe chest recessions | 3.01 | 1.34–6.37 | 0.005* |
| Temperature instability >38˚C or <35.5˚C | 1.70 | 0.75–3.75 | 0.191 |
| Central cyanosis | 4.15 | 1.62–9.87 | 0.002* |

*Significant ORs (p<0.05)

**Table 3. Association between DS and bacterial sepsis when organism is a known pathogen.**

| Variable | Odds ratio (OR) | 95%CI | P value |
|---|---|---|---|
| Not feeding well | 2.25 | 1.17–4.44 | 0.017* |
| Convulsions | 0.65 | 0.10–2.30 | 0.571 |
| Drowsy / Difficult to wake | 0.86 | 0.20–2.57 | 0.811 |
| Movements only when stimulated or no movements | 1.65 | 0.46–4.59 | 0.382 |
| Fast breathing > = 60 per min | 1.06 | 0.49–2.17 | 0.868 |
| Grunting | 2.38 | 0.84–5.89 | 0.077 |
| Severe chest recessions | 2.17 | 0.92–4.74 | 0.061 |
| Temperature instability >38˚C or <35.5˚C | 1.60 | 0.78–3.24 | 0.193 |
| Central cyanosis | 1.01 | 0.32–3.03 | 0.994 |

*Significant ORs (p<0.05)

Mortality was higher in infants with bacterial sepsis with a known neonatal pathogen compared to other infants (22.0% [9/41] versus 8.3% [30/360]; p = 0.005). Bacterial sepsis was significantly associated with mortality (OR 3.07 [95%CI: 1.28–6.82]; p = 0.008). However, the only DS significantly associated with bacterial sepsis when considering only known neonatal pathogens was *not feeding well* (OR 2.25 [95%CI: 1.17–4.44]; p = 0.017 (Table 3); 61.0% [25/41] of the children with bacterial sepsis were not feeding well versus 41.2% [146/354] of the other children). Associations between convulsions, drowsiness, lethargy (i.e., movements only when stimulated or no movement), tachypnea, grunting, severe chest recession, temperature instability and central cyanosis were all non-significant (Table 3). Association was between cumulative number of DS and bacterial sepsis was marginally signiSficant (OR 1.22 [95% CI: 1.00–1.49]; p = 0.046) (S2 Fig). At discharge, clinical staff selected non-bacterial sepsis in 76.3% (306/401), bacterial sepsis in 15.5% (62/401) and "other non-sepsis syndromes" in 8.2% (33/401) of cases, from these options, for study purposes.

## Associations when including potential contaminant blood cultures

When considering positive blood cultures for potential contaminants (S1 Table), an additional 45 infants had positive blood or CSF cultures (total of 86 infants with positive blood cultures). One infant had a contaminating blood culture, but grew unidentified gram-negative diplococci in the CSF, so this infants was originally classified as bacterial sepsis with a pathogen (total of 46 infants with potential blood culture contaminants). When including infants with positive blood cultures for both contaminants and pathogens, and infants with positive CSF cultures, culture positivity was not significantly associated with mortality (OR 1.74 [95% CI: 0.82–3.53]; p = 0.135), or any DS (S3 Table). The number of DS was also not significantly associated with bacterial sepsis when including all pathogens and potential contaminants (OR 1.09 [95% CI: 0.93–1.28]; p = 0.273; S3 Fig).

## Discussion

This study found that most WHO DS were significantly associated with mortality, but only one DS (i.e., not feeding well) as significantly associated with bacterial sepsis with a positive blood or CSF culture for a neonatal pathogen. Additionally, none of the other DS were significantly associated with bacterial sepsis in this LMIC neonatal cohort. The latter results held true whether only bacterial pathogens, or pathogens plus potential bacterial contaminants were considered. Notably, a significant minority (39%) of parent of infants with bacterial sepsis initially declared that their child was feeding well at presentation. Therefore, absence of this DS could practically miss a large proportion of infants with bacterial sepsis in real life. Few studies

have attempted to determine how DS can predict bacterial sepsis caused by bacterial pathogens in young infants. A retrospective Brazilian study reviewed data on 83 sick neonates over a 12-month period, and came to the same conclusion that DS alone or in combination did not predict severe serious bacterial infection [15]. The current study findings also support findings from another study carried out in primary care, rural setting in Bangladesh, India and Pakistan [16], clinical signs could not differentiate infants with viral or bacterial illnesses. To the best of our knowledge, however, the current study is the first to pragmatically examine whether DS predict bacterial sepsis in infants. These findings were obtained in an LMIC setting where DS were prospectively identified by front line healthcare workers as they would normally do in real-life, outside a research protocol–we assumed that this study design would more closely reflects the practical utility of DS. The current study findings have implications for a hospital setting. Essentially, it suggests that DS cannot reliably identify infants with bacterial sepsis who require antibiotic treatment. However, further research is needed to determine the generalizability of the results to other hospital contexts.

The strength of this study are its robust standardized blood culture outcome measures where contamination during blood sampling were minimized by using protocolized aseptic procedures, and the robust microbiological identification methods, including access to a certified clinical laboratory lab and protocols on-site augmented by advanced MALDI-TOF species determination in another, North American clinical microbiology laboratory. The strong association between bacterial sepsis and mortality provides reassurance infants with positive cultures were correctly identified.

In addition to the main findings, the current study also found that only a minor proportion of infants with suspected sepsis showed positive blood culture. A limited number of studies have prospectively estimated the incidence of bacterial sepsis in LMIC cohorts, and can be discussed here in the context of the current study findings. A hospital-centred study in Delhi, India, included 13,530 neonates (all comers; mean GA±SD: 36.0±3.4 weeks) monitored daily for clinical signs of sepsis with per-protocol completion of a sepsis work-up (blood culture and lumbar puncture) prior to initiating antibiotics, and found a 14.3% incidence of culture-positive sepsis [17], which is in keeping with the current study findings. Another study in community settings in Madagascar, Senegal and Cambodia reported 8.2% culture-confirmed sepsis among 514 neonates assessed for "possible serious bacterial infection" out of 3,688 infants followed during the neonatal period [18]. One large study followed 12,622 live births up to 60 days of age in rural India between April 2002 and March 2005, with daily home visits and clinical assessment, and standardized blood and CSF cultures following enrollment of infants with suspected sepsis. It reported a 6.3% incidence of culture-confirmed sepsis (53 with positive blood cultures out of 842 infants assessed, excluding potential contaminants), for a population-level incidence of culture-confirmed sepsis of 6.7 per 1,000 births [19]. Taken together, these findings support incidence rates found in the current study, and provide robust data on the incidence of bacterial sepsis in LMIC settings, and form the basis for future studies to measure the impact of potential interventions.

Blood cultures are the gold standard for diagnosing bacterial sepsis in young infants [20]. However, these tests lack sensitivity in neonates [21]. In the current study we were careful to prescribe a minimum blood volume to maximize culture yield. Yet, a key question is whether we misestimate the proportion of bacterial sepsis in infants in whom cultures are negative? The Aetiology of Neonatal Infection in South Asia (ANISA) study [22] also attempted to identify the cause of sepsis for 6022 cases of possible serious bacterial infections in infants in the first 60 days after birth, in a population of 63,114 livebirth). Bacterial and viral (e.g., respiratory syncytial virus) causes were identified in 16% and 12% of cases, respectively. For the remaining 72% cases, no pathogen could be identified despite systematic blood and respiratory cultures, which is consistent with findings from the current study.

Evidently the use of antibiotics in the current study was high, with the vast majority of infants receiving antibiotics for a minimum of five days during hospitalization, as per hospital practices. As in other LMIC settings, blood cultures were not routinely available at the study site prior to the study. As such, no approach had been developed to stop antibiotics after 48 hours given a negative blood culture. In another study in Thailand, the duration of antibiotic use was also high with durations of 15 and 8 days for culture-positive and culture-negative early-onset neonatal sepsis, respectively [23]. These results highlight the limits of having to rely on DS, and potential coercive effect of the lack of laboratory resources capable of providing basic laboratory test results to help "rule out" sepsis [24].

A number of study limitations should be acknowledged. Although a sizeable proportion of the recruited infants presented from home, data were collected in a single hospital-type setting; as such, caution should be exercised when generalizing to other hospital settings or extrapolating to non-hospital primary care settings. For logistical reasons CSF was collected from only approximately half of the infants, which may have underestimated the incidence of bacterial sepsis. Similarly, urine or anaerobic blood cultures were not collected, which may have resulted in missed bacteremia. Some infants received antibiotics prior to blood culture, which may have lowered bacterial sepsis estimates. Moreover, due to the limited diagnostic capabilities of the study LMIC settings, in most cases, causes of deaths or alternate diagnoses could not be determined. Similarly, the study did not include non-bacterial testing, which precluded identification of viral sepsis syndromes.

In conclusion, this study validates the use of WHO DS to predict fatal outcomes, but not to reliably identify infants with bacterial sepsis. In other words, relying solely on DS to identify infants with bacterial sepsis is likely to lead to overtreatment while at the same time missing infants who require antibiotic treatment. This study further emphasizes the need for better diagnostics for bacterial sepsis, to ensure that health gains continue to improve, in areas of the world where sepsis represents a huge burden and where antimicrobial resistance is also increasing at alarming rates [25].

## Supporting information

**S1 Table. Bacterial isolates categorized as pathogens or potential contaminants in secondary analysis using the modified bacterial sepsis outcome.** *Bacteria were also cultured from cerebrospinal fluid (CSF).
(DOCX)

**S2 Table. Baseline infant characteristics associated with increased mortality.** £adjusted OR for age of mother, gestational age, birth and admission weights, age of the infant and sex; *Significant ORs (p<0.05).
(DOCX)

**S3 Table. Associations between DS and bacterial sepsis when all organisms are included (i.e., including both pathogens and potential contaminants).**
(DOCX)

**S1 Fig. Relationship between DS and mortality (with corresponding data).** Plots of model fits—Death as outcome.
(DOCX)

**S2 Fig. Relationship between DS and pathogen bacterial sepsis (with corresponding data).** Plots of model fits—pathogen sepsis as outcome.
(DOCX)

**S3 Fig. Relationship between DS and bacterial sepsis including contaminants (with corresponding data).** Plots of model fits–All sepsis as outcome.
(DOCX)

**S1 Questionnaire. PLOS inclusivity in global research questionnaire.**
(DOCX)

**S1 Dataset. Raw, line-level dataset (Excel spreadsheet) for the study and cohort.**
(XLSX)

**S1 File. STROBE checklist.**
(DOCX)

## Acknowledgments

We thank Bentry Tembo, Rhoda Chifisi and Blessings Chiluzi for help with recruitment and data collection, Charles Mwansambo, Patience Johane and the staff at the PACHI for administrative and operational support during the study, Robert Krysiak, Gerald Tegha and the staff of the University of North Carolina–Malawi lab for performing all bacterial cultures and identification, on-site, in Lilongwe, according to clinical laboratory standards, Dr. David Goldfarb for expert input into data interpretation, Amber Johnson from the British Columbia Children's Hospital Microbiology Laboratory for secondary confirming bacterial isolates by mass spectrometry.

## Author Contributions

**Conceptualization:** Omolabake Akinseye, Pascal M. Lavoie.

**Data curation:** Omolabake Akinseye, Constantin R. Popescu, Msandeni Chiume-Kayuni, Norman Lufesi.

**Formal analysis:** Constantin R. Popescu, Michael A. Irvine.

**Funding acquisition:** Msandeni Chiume-Kayuni, Norman Lufesi, Pascal M. Lavoie.

**Investigation:** Tisungane Mvalo, Niranjan Kissoon, Matthew O. Wiens, Pascal M. Lavoie.

**Methodology:** Michael A. Irvine, Pascal M. Lavoie.

**Project administration:** Norman Lufesi.

**Resources:** Norman Lufesi.

**Supervision:** Msandeni Chiume-Kayuni, Norman Lufesi, Pascal M. Lavoie.

**Validation:** Constantin R. Popescu.

**Writing – original draft:** Omolabake Akinseye, Pascal M. Lavoie.

**Writing – review & editing:** Constantin R. Popescu, Msandeni Chiume-Kayuni, Michael A. Irvine, Norman Lufesi, Tisungane Mvalo, Niranjan Kissoon, Matthew O. Wiens.

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
