## [Decision Letter · Decision Letter 0]

29 Jun 2023

PGPH-D-23-00809

World Health Organization Danger Signs to predict bacterial sepsis in newborns: A pragmatic prospective cohort study

Dear Dr. Lavoie,

Thank you for submitting your manuscript to PLOS Global Public Health. After careful consideration, we feel that it has merit but does not fully meet PLOS Global Public Health’s publication criteria as it currently stands. Therefore, we invite you to submit a revised version of the manuscript that addresses the points raised during the review process.

Please note that we have only been able to secure a single reviewer to assess your manuscript. We are issuing a decision on your manuscript at this point to prevent further delays in the evaluation of your manuscript. Please be aware that the editor who handles your revised manuscript might find it necessary to invite additional reviewers to assess this work once the revised manuscript is submitted. However, we will aim to proceed on the basis of this single review if possible.

Your manuscript has been assessed by an expert reviewer, whose full comments are appended below. The reviewer has requested clarification on several aspects of the methodology and interpretation of the findings. Please ensure you respond to each point carefully in your response to reviewers document, and modify your manuscript accordingly.

We look forward to receiving your revised manuscript.

Kind regards,

Dr Joseph Donlan

Editorial Office

Journal Requirements:

2. We noticed that you used "not shown" in the manuscript. We do not allow these references, as the PLOS data access policy requires that all data be either published with the manuscript or made available in a publicly accessible database. Please amend the supplementary material to include the referenced data or remove the references.

3. We have noticed that you have a list of Supporting Information legends in your manuscript. However, there are no corresponding files uploaded to the submission. Please upload them as separate files with the item type 'Supporting Information'. 

Additional Editor Comments (if provided):

Reviewers' comments:

Reviewer's Responses to Questions

**Comments to the Author**

1. Does this manuscript meet PLOS Global Public Health’s publication criteria? Is the manuscript technically sound, and do the data support the conclusions? The manuscript must describe methodologically and ethically rigorous research with conclusions that are appropriately drawn based on the data presented.

Reviewer #1: Partly

2. Has the statistical analysis been performed appropriately and rigorously?

Reviewer #1: Yes

3. Have the authors made all data underlying the findings in their manuscript fully available (please refer to the Data Availability Statement at the start of the manuscript PDF file)?

Reviewer #1: Yes

4. Is the manuscript presented in an intelligible fashion and written in standard English?

Reviewer #1: Yes

5. Review Comments to the Author

Reviewer #1: The manuscript is written in such a way that it is easy to understand. The study topic is interesting, sufficient justification is provided in the background. This study revealed that most WHO danger signs are not a predictor of neonatal bacterial sepsis, even if in some circumstances, antibacterial treatment is initiated based on the presence of these signs. This study also indicates that a substantial portion of sepsis-suspected newborns have negative blood cultures or that causative agents are missed because of the inherent problems of the diagnostic method. The concern is WHO danger signs and clinical presentation of bacterial sepsis may overlap. Dealing with ‘danger signs’ and ‘neonatal sepsis’ without discussing/mentioning the clinical picture of neonatal sepsis makes things unclear for the reader. As described in the manuscript, the entry point for the study participant was ‘suspected of sepsis’ and not ‘showing danger signs’. It is good if an attempt was made to describe this issue in the manuscript. What does this study recommend? Is bacterial sepsis missed because of the low sensitivity of the diagnostic method used? Or are there other causes? In case of blood culture-negative results where there are danger signs, is it appropriate not to give antibiotics? It is clear that the use of antibiotics in the absence of bacterial etiology has its own disadvantages; however, the risk and benefits should be sought. All these issues need to be reflected in the discussion. A clear implication of the study has to be stated.

General comment

• In the main method give the operation definition ‘suspected sepsis’ in newborns.

• Which types of bacteria were isolated from CSF?

• Some terms are inconsistent within the manuscript “young infants”, “Newborns”, “Neonates”, “infants” It is good to define the study population and use it consistently throughout the manuscript.

• The study design, sample size (500 vs 401 vs) is not clearly mentioned.

• Consider replacing ‘h’ with ‘hours’ if it means so

• Line # 144 “enrollment” Line # “approximatively” check the spelling

• Mention the nine WHO danger signs in the clinical data and samples collection section.

• Was the attempt made to isolate anaerobic bacteria? If not. include in the limitation

• As the etiology of neonatal sepsis varies with age; I suggest considering presenting the result section by taking into account the age classification of newborns (< 7 days of age and 7-90 days of age)

• What were the criteria to collect CSF?

• How bacteria were identified after blood culture is not clear, for how long the blood culture was incubated before it was declared negative.

• There is no description of how CSF was cultured

Specific comments

• In the abstract section, I suggest separating methods and findings/results

• In the abstract it is written as “This prospective study evaluated nine DS in infants younger than 3 months…” Do newborns with the age of 3 months were included?

• Line #93: “Over time, the use of DS has been extended to provide a reasonable basis for the initiation of empiric antibiotic treatment (9). However, these signs have neither been developed nor validated to diagnose sepsis of bacterial etiology.” Is it correct to use reference #9 for the information given? Reference #9 looks like a guideline developed by WHO, is the WHO recommending the use of empiric antibiotic treatment without evidence?

• In material and method section line #109 “study that aimed to identify molecular signatures of bacterial sepsis in young infants…” was molecular techniques used to diagnose sepsis?

• Line #113-114: “…. infants were excluded if they had received antibiotics for >4 hours at the time of enrollment.” What if they have received antibiotics treatment in the last two weeks?

• Line # 114-117: Out of 851, how many refused to participate, the condition was not permissive, and how many were included in the study?

• Line #119: “The Kamuzu Central Hospital the largest referral hospital in central Malawi, where approximatively 3,000 infants are admitted to its neonatal unit annually” needs reference

• Line # 128-129: “During the 129 study, the authors did not have access to information that could identify participants…” if this is the case how lab results were communicated to the attending physician? Line # 150 “Culture results were made available to clinicians within 5 days.” And also the recommended incubation period of blood culture is about a week, how culture result is available for the physician within 5 days?

• Line # 193-195: “In total, 94/401 (23.4%) infants had received antibiotics prior to the blood culture, for a median of 2 hours (range 0.75 to 3.5 195 hours)” This could have affected the findings

• Some data were collected from the baby’s mother for example gestational age, and mode of delivery. Was consent sought from mothers?

• Table 1 needs to be improved. Why values in the column ‘N with data’ are variable given the total sample size is 401? Fifth row what does ‘inborn’ mean?

• Table 1: Why capturing “Illness status, number on infants” is needed provided that the study population are newborns suspected of sepsis (ill) visiting the hospital

• Line # 202: The contamination rate is somewhat high “However, about half (46 / 85) had bacterial types considered to be contaminants” If the blood specimen were collected aseptically they may be true pathogens.

• Line 203-204: “…and non-pathogenic streptococcus species such as oralis)…” you may rewrite as ‘….and non-pathogenic Streptococcus species such as S. oralis)…’ Consider scientific name during writing the name of bacteria.

• Table 2: it is good if a column is added to show frequency and proportion in the ‘yes’ and ‘No’ categories for all variables. Doing so will make the reader see the prevalence in each cell.

Danger signs Mortality, n(%) OR 95%CI p-value

Yes No

Not feeding well N(%) N(%)

Convulsion N(%) N(%)

• Table 3: it is good if a column is added to show frequency and proportion in yes and No category for all variables.

Danger signs Blood culture result, n(%) OR 95%CI p-value

Yes No

Not feeding well N(%) N(%)

Convulsion N(%) N(%)

• Tables 2 and 3: I don’t see the importance of using (), it can be omitted.

• Discussion lines # 277-278: “Taken together, these findings suggest that bacterial sepsis may be uncommon in this age group even in LMICs when sepsis is clinically suspected.” This is an area of controversy in LMICs and sub-Saharan African countries. Proxy studies and systematic reviews and meta-analyses reported a high burden of neonatal sepsis in these regions. Strong recommendations should be forwarded to concerned bodies so that similar studies (like the present study) using robust diagnostic methods to be conducted in the region. This will assist in device appropriate prevention strategies for the regions which may include vaccines and measure the impact of the prevention strategy. Without knowing the real burden of neonatal sepsis along with its etiology we may not objectively measure the impact of the prevention strategy to be used.

• Discussion: line# 302-304: “The strong association between bacterial sepsis and mortality suggest that the infants with positive cultures were correctly identified” Is there a finding which supports this claim?

6. PLOS authors have the option to publish the peer review history of their article (what does this mean?). If published, this will include your full peer review and any attached files.

**Do you want your identity to be public for this peer review?** For information about this choice, including consent withdrawal, please see our Privacy Policy.

Reviewer #1: **Yes: **Musa Mohammed Ali

---

## [Decision Letter · Decision Letter 1]

26 Sep 2023

PGPH-D-23-00809R1

World Health Organization Danger Signs to predict bacterial sepsis in young infants: A pragmatic cohort study

Dear Dr. Lavoie,

Thank you for submitting your manuscript to PLOS Global Public Health. After careful consideration, we feel that it has merit but does not fully meet PLOS Global Public Health’s publication criteria as it currently stands. Therefore, we invite you to submit a revised version of the manuscript that addresses the points raised during the review process.

We invite you to address all the comments given by reviewer including the selection criteria to select study participants, sample size was determination and contamination rate, .

We look forward to receiving your revised manuscript.

Kind regards,

Musa Mohammed Ali, PhD

Guest Editor

Journal Requirements:

1. “Please include a complete copy of PLOS’ questionnaire on inclusivity in global research in your revised manuscript. Our policy for research in this area aims to improve transparency in the reporting of research performed outside of researchers’ own country or community. The policy applies to researchers who have travelled to a different country to conduct research, research with Indigenous populations or their lands, and research on cultural artefacts. The questionnaire can also be requested at the journal’s discretion for any other submissions, even if these conditions are not met.  Please find more information on the policy and a link to download a blank copy of the questionnaire here: https://journals.plos.org/globalpublichealth/s/best-practices-in-research-reporting. Please upload a completed version of your questionnaire as Supporting Information when you resubmit your manuscript.

Additional Editor Comments (if provided):

Reviewers' comments:

Reviewer's Responses to Questions

**Comments to the Author**

1. If the authors have adequately addressed your comments raised in a previous round of review and you feel that this manuscript is now acceptable for publication, you may indicate that here to bypass the “Comments to the Author” section, enter your conflict of interest statement in the “Confidential to Editor” section, and submit your "Accept" recommendation.

Reviewer #2: All comments have been addressed

2. Does this manuscript meet PLOS Global Public Health’s publication criteria? Is the manuscript technically sound, and do the data support the conclusions? The manuscript must describe methodologically and ethically rigorous research with conclusions that are appropriately drawn based on the data presented.

Reviewer #2: Yes

3. Has the statistical analysis been performed appropriately and rigorously?

Reviewer #2: Yes

4. Have the authors made all data underlying the findings in their manuscript fully available (please refer to the Data Availability Statement at the start of the manuscript PDF file)?

Reviewer #2: Yes

5. Is the manuscript presented in an intelligible fashion and written in standard English?

Reviewer #2: Yes

6. Review Comments to the Author

Reviewer #2: The overall study concepts are very important especially from the study setting point of view.

I had the following comments and point of clarification for author.

#1. How were the study participants selected? What was the criteria included in the study? How was sepsis suspected was defined for this study?

#2. Why high figure of non-response rate (401/851 approached)? Was it the problem of awareness or high price of blood culture? How do you explain this for researchers.

#3. Clarify how you had determined the sample size in your plan (500) and achievement (400).

#4. In page 6 line 155 you mentioned about urine culture in your work but can’t found any in the finding or method related with this. Clarify!

#5. You did CSF culture for the selected infants in the study. How did you select for those who CSF culture done and not done?

#6. Most of children was on antibiotics for the median?? Of 6 days. How this matched with your mention exclusion criteria, more than 4hrs? May the BACTEC have an advantage of reducing the effect of antibiotics on bacterial growth?

#7. How were contaminants defined in the study? Have you done with more than one bottle blood culture? Is there a possibility for those bacteria like CoNS, to be a potential pathogen?

#8. What is the total contaminant rate in the study based on your definition used? How this rate comparable with the tolerable WHO blood culture rate? Figure on the supplementary table is high!

#9. Once you have an identified bacterium, why can’t look for antibiotic resistance?

#10. What is the overall about the conclusion of your study? Does it conclude with DS more related with sepsis caused by other than bacterial infections or what? elaborate?

7. PLOS authors have the option to publish the peer review history of their article (what does this mean?). If published, this will include your full peer review and any attached files.

**Do you want your identity to be public for this peer review?** For information about this choice, including consent withdrawal, please see our Privacy Policy.

Reviewer #2: No

---

## [Decision Letter · Decision Letter 2]

18 Oct 2023

World Health Organization Danger Signs to predict bacterial sepsis in young infants: A pragmatic cohort study

PGPH-D-23-00809R2

Dear Dr Lavoie,

We are pleased to inform you that your manuscript 'World Health Organization Danger Signs to predict bacterial sepsis in young infants: A pragmatic cohort study' has been provisionally accepted for publication in PLOS Global Public Health.

Best regards,

Musa Mohammed Ali, PhD

Guest Editor

Reviewer Comments (if any, and for reference):

Reviewer's Responses to Questions

**Comments to the Author**

1. If the authors have adequately addressed your comments raised in a previous round of review and you feel that this manuscript is now acceptable for publication, you may indicate that here to bypass the “Comments to the Author” section, enter your conflict of interest statement in the “Confidential to Editor” section, and submit your "Accept" recommendation.

Reviewer #1: All comments have been addressed

Reviewer #2: All comments have been addressed

2. Does this manuscript meet PLOS Global Public Health’s publication criteria? Is the manuscript technically sound, and do the data support the conclusions? The manuscript must describe methodologically and ethically rigorous research with conclusions that are appropriately drawn based on the data presented.

Reviewer #1: Yes

Reviewer #2: Yes

3. Has the statistical analysis been performed appropriately and rigorously?

Reviewer #1: Yes

Reviewer #2: Yes

4. Have the authors made all data underlying the findings in their manuscript fully available (please refer to the Data Availability Statement at the start of the manuscript PDF file)?

Reviewer #1: Yes

Reviewer #2: Yes

5. Is the manuscript presented in an intelligible fashion and written in standard English?

Reviewer #1: Yes

Reviewer #2: Yes

6. Review Comments to the Author

Reviewer #1: All comments are addressed.

Reviewer #2: Authors address comments raised during the review process!

7. PLOS authors have the option to publish the peer review history of their article (what does this mean?). If published, this will include your full peer review and any attached files.

**Do you want your identity to be public for this peer review?** For information about this choice, including consent withdrawal, please see our Privacy Policy.

Reviewer #1: No

Reviewer #2: **Yes: **Adugna Negussie Gudeta
